# Distinct UPR and Autophagic Functions Define Cell-Specific Responses to Proteotoxic Stress in Microglial and Neuronal Cell Lines

**DOI:** 10.3390/cells13242069

**Published:** 2024-12-15

**Authors:** Helena Domínguez-Martín, Elena Gavilán, Celia Parrado, Miguel A. Burguillos, Paula Daza, Diego Ruano

**Affiliations:** 1Departamento de Bioquímica y Biología Molecular, Facultad de Farmacia, Universidad de Sevilla (US), 41012 Sevilla, Spain; hamaralv@gmail.com (H.D.-M.); egavilan@us.es (E.G.); parradocelia@gmail.com (C.P.); maburguillos@us.es (M.A.B.); 2Instituto de Biomedicina de Sevilla (IBIS), Hospital Universitario Virgen del Rocío/Consejo Superior de Investigaciones Científicas (CSIC)/Universidad de Sevilla (US), 41013 Sevilla, Spain; 3Departamento de Biología Celular, Facultad de Biología, Universidad de Sevilla (US), 41012 Sevilla, Spain; pdaza@us.es

**Keywords:** proteotoxic stress, autophagy, proteasome, microglia, neurons, phagocytosis, proteostasis

## Abstract

Autophagy is a catabolic process involved in different cellular functions. However, the molecular pathways governing its potential roles in different cell types remain poorly understood. We investigated the role of autophagy in the context of proteotoxic stress in two central nervous system cell types: the microglia-like cell line BV2 and the neuronal-like cell line N2a. Proteotoxic stress, induced by proteasome inhibition, produced early apoptosis in BV2 cells, due in part to a predominant activation of the PERK-CHOP pathway. In contrast, N2a cells showcased greater resistance and robust induction of the IRE1α-sXbp1 arm of the UPR. We also demonstrated that proteotoxic stress activated autophagy in both cell lines but with different kinetics and cellular functions. In N2a cells, autophagy restored cellular proteostasis, while in BV2 cells, it participated in regulating phagocytosis. Finally, proteotoxic stress predominantly activated the mTORC2-AKT-FOXO1-β-catenin pathway in BV2 cells, while N2a cells preferentially induced the PDK1-AKT-FOXO3 axis. Collectively, our findings suggest that proteotoxic stress triggers cell-specific responses in microglia and neurons, with different physiological outcomes.

## 1. Introduction

Maintaining cellular proteostasis, the balance between synthesis and protein degradation, is crucial for cell survival [1,2]. Protein degradation is regulated by the two major proteolytic systems: the ubiquitin-proteasome system (UPS) and macroautophagy (hereafter referred to as autophagy) [3,4,5]. A growing body of evidence demonstrates that the functional cooperation between these two systems, particularly autophagy compensating for proteasome dysfunction, is fundamental for cell survival [6,7,8]. This issue is particularly relevant during aging, as the loss of functional cooperation renders neurons more vulnerable to various stress situations [9,10]. Proteotoxic stress constitutes a significant challenge, especially for post-mitotic cells like neurons, where age-dependent protein accumulation underlies various neurodegenerative disorders. In the central nervous system (CNS), both UPS and autophagy have been extensively studied in neurons, highlighting their critical role in promoting cellular health and neuronal survival [9,11,12,13]. However, in microglial cells, limited data are available regarding their functional relationships, as well as the physiological role of autophagy and the signaling pathways regulating this process under proteotoxic stress [14,15,16]. Microglia, as the resident macrophage-like cells of the CNS, are involved in various functions such as synaptic organization, neuronal circuit plasticity, inflammatory response, neuronal support, and phagocytosis [17].

Phagocytic microglia can engulf and degrade diverse products, including myelin, deposits of aggregated proteins, microbes, and dead cells [18]. They can also phagocytize themselves when undergoing apoptosis or necrosis, a process known as microglial cannibalism [19,20]. Furthermore, recent studies have shown that the endocytosis of extracellular vesicles stimulates autophagy in microglia [21]. However, the signaling pathways regulating microglial phagocytic activity in different scenarios remain poorly understood. Multiple studies suggest that autophagy and phagocytosis may be two complementary and interdependent processes in both macrophages [22,23] and microglial cells [24,25,26,27]. In this study, we conducted a comparative analysis of the cellular responses induced by proteotoxic stress in the murine microglia-like cell line BV2 and the neuronal-like cell line N2a. We specifically focused on the activation of the UPR and autophagy. Our key findings indicate cell-specific responses to proteotoxic stress, involving the activation of the UPR and the physiological role of autophagy.

## 2. Materials and Methods

### 2.1. Cell Lines and Culture Conditions

This study was performed in the BV2 murine microglial cell line and the N2a murine neuronal cell line. The BV2 cell line was cultured in DMEM/high-glucose medium, and the N2a cell line in DMEM/high-glucose/OptiMEM (50–50%). Both lines were supplemented with 2 mM glutamine and 10% fetal bovine serum in the presence of penicillin (100 units/mL) and streptomycin (0.01 mg/mL). All from Thermo Fisher Scientific (Madrid, Spain). The cells were maintained at 37 °C in a humidified incubator with an atmosphere of 5% CO_2_.

### 2.2. Treatments and Drugs

Both cell lines were treated in parallel and at indicated times with the following chemical compounds: 1 μM of MG132 (reversible proteasome inhibitor; Merck Life Science, Madrid, Spain); 5 mM of 3-MA (autophagy inhibitor); 20 μM of STF-083010 (IRE1α pathway inhibitor); 5 μM of GSK2606414 (PERK pathway inhibitor); 20 μM of GSK-3β inhibitor VII; 1 µM of bafilomycin (selective inhibitor of vacuolar-type H + ATPase. All reagent from Merck Life Science, Madrid, Spain). MG132, bafilomycin, STF-083010, GSK2606414, and GSK-3β inhibitor VII were dissolved in DMSO. The same amount of water or DMSO used for treatments was also used for control cells. The volume of DMSO was less than 1% of the final volume, and it was not toxic to the cells. The control groups were exposed for the maximum duration of treatment and treated with the highest concentrations of the vehicles used for dissolving the treatment compounds. All samples, including both control and treated groups, were collected simultaneously at the conclusion of the treatment period to ensure uniform handling and minimize variability in processing.

### 2.3. Analysis of Apoptosis by Flow Cytometry

Treatments with MG132 were conducted without FBS. Staurosporine (Sigma-Aldrich) at 0.5 μg/mL served as a positive apoptosis control. Cell viability was assessed using the Annexin V-FITC Apoptosis Detection Kit (Immunostep) following the manufacturer’s instructions and the BD FACS Canto II flow cytometer. Results were analyzed using FACSDivaTM software v. 6.1.3 (BD Bioscience). This kit utilizes Annexin V-FITC, which binds to exposed phosphatidylserine during apoptosis, and propidium iodide (PI) to assess membrane integrity. Simultaneous staining with Annexin V and PI discriminates viable cells (Annexin V-/PI-), early apoptotic cells (Annexin V+/PI-), late apoptotic cells and necrotic cells (Annexin V+/PI+).

### 2.4. Acridine Orange Staining

Acidic compartments were visualized using acridine orange staining. Cells were seeded on coverslips in the absence or the presence of MG132 (1 μM, for 6 h). Cells were then washed with PBS and incubated for 10 min with a medium containing 0.1 mg/mL acridine orange (Molecular Probes, Alcobendas, Spain). Then coverslips were washed with PBS for 5 min three times to remove the acridine orange. Fluorescent images were taken using an inverted fluorescent microscope (Leica, Weztlar, Germany).

### 2.5. Western Blotting

Immunoblots were performed as previously described [7]. Total proteins were extracted using the TRIsureTM isolation reagent (BIO-38033, Bioline). Protein pellets were solubilized in 4% SDS–8M urea. Once quantified by Lowry assay, proteins in Laemmli buffer were loaded on a 12 or 14% polyacrylamide gel for electrophoresis (SDS-PAGE; Bio-Rad, Alcobendas, Spain) and then transferred to a nitrocellulose membrane (Hybond-C Extra; Amersham, Barcelona, Spain). Membranes were blocked and then incubated overnight at 4 °C with the following primary antibodies: (i) rabbit polyclonal antibodies against Akt, phospho-Akt (T308), phospho-Akt (S473), cleaved caspase-3, LC3B, phospho-GSK-3β (S9), GRP78, poly-ubiquitin, SQSTM1/p62, and phospho-SQSTM1/p62 (S403), phospho-FOXO3 (S253), and phospho-FOXO1 (S256) (Cell Signaling, Danvers, MA, USA); XBP1 (abcam, Cambridge, UK); (ii) mouse monoclonal antibodies against β-actin (Sigma-Aldrich), β-catenin, and phospho-β-catenin (S33/37/T41), CHOP, FOXO3, FOXO1 (Cell Signaling, Danvers, MA, USA); GSK3-clone 4G-1E (Millipore, Madrid, Spain). Membranes were incubated with the corresponding secondary antibodies (Dako) horseradish peroxidase-conjugated and developed using the ECL-plus detection method (Amersham) and the ImageQuant LAS 4000 MINI GOLD (GE Healthcare Life Sciences, Barcelona, Spain). The optical density of individual bands was analyzed using the ImageJ software and normalized relative to the optical density of the housekeeping β-actin.

### 2.6. RNA Extraction and Reverse Transcription

For PCR analysis, total RNA was extracted using the TRIsureTM isolation reagent (BIO-38033, Bioline), following the manufacturer’s instructions. The recovery of RNA was similar in both cell lines. For reverse transcription experiments, we used the high-capacity reverse transcription kit (Thermofisher, Madrid, Spain), according to the manufacturer’s instructions.

### 2.7. Real-Time PCR

For PCR amplification, the cDNAs were diluted (1/10) in sterile water and used as a template as previously described in [28]. Real-time PCR was performed in the ABI Prism 7000 sequence detector (Applied Biosystems, Barcelona, Spain), using TaqMan probes designed by Applied Biosystems or specifically designed primers for the spliced variant of the xbp1 gene (from 5′ to 3′ UP: CTTGTGGTTGAGAACCAGG; and LO: GGCCTGCACCTGCTGCGGACTC). The expression levels of the different genes were normalized by the expression of two different housekeepers, GAPDH (glyceraldehyde-3-phosphate dehydrogenase) and β-actin. The amplification of the housekeepers was performed in parallel with the genes to be analyzed. Similar results were obtained using both housekeepers. Threshold cycle (Ct) values were calculated using the software supplied by Applied Biosystems.

### 2.8. Preparation of Cell Debris and Phagocytosis Assay

For cell debris preparation, BV2 cells were stained with 50 μM of 5-TAMRA fluorescent dye (5-Carboxytetramethylrhodamine, ab145438 Abcam) for 30 min at RT. Cells were then washed 3 times with PBS and harvested with a cell scraper. The collected pellet was washed 3 times to remove any amount of free TAMRA and finally resuspended in PBS (107 cells per mL of PBS). Cells were then disrupted with a 25-gauge needle syringe. To test whether the cell debris can be engulfed by cultured BV2 cells, 100 μL of cell debris (corresponding to debris from 106 cells) was added to each well of a 6-well plate. Together with cell debris, cells were treated with MG132 1 μM, 3-MA 1 mM, or the combined treatment for 6 h. The 3-MA compound was added 1 h before treatments. To distinguish the cultured cells and non-disrupted cells from debris, cultured cells were previously stained with Hoechst (1:10000) and washed before adding the cell debris. Finally, the cells were scraped, washed with PBS, and fixed with 4% PFA for 15 min at RT and washed again and resuspended in PBS. The engulfment of debris was then analyzed by cytometry using the CytoFlex S (Beckman Coulter). Hoechst was analyzed with the violet (405 nm) laser and the PB450 channel. TAMRA was analyzed with the blue (488 nm) laser and PE channel. Phagocytic cells were positive for both PB450 and PE fluorescences.

### 2.9. Statistical Analysis

Data are expressed as means ± S.D. and were analyzed by one-way ANOVA followed by Bonferroni’s post hoc test using the program Statgraphics Plus. Differences were considered significant at *p*  <  0.05. Experiments were repeated 3 to 5 times.

## 3. Results

### 3.1. BV2 Cells Were More Sensitive than N2a Cells to Proteotoxic Stress

We initially assessed the impact of proteasome inhibition on cell viability in both cell lines. To achieve this, we utilized the reversible proteasome inhibitor MG132 and examined cell viability through flow cytometry. The data indicated that N2a cells exhibited greater resistance to proteotoxic stress compared with BV2 cells (Figure 1A). Following a two-hour incubation with MG132, BV2 cells demonstrated a significant time-dependent decrease in cell viability. In contrast, N2a cell viability remained stable and was only slightly but significantly reduced from 6 to 8 h (Figure 1B). To further characterize the type of cellular death, we analyzed the expression of processed caspase-3. Processed fragments of caspase-3 were observed in BV2 cells from 4 to 8 h after proteasome inhibition but were not detected in N2a cells under these conditions (Figure 1C). Thus, cytometry and biochemical data indicate that our model of proteotoxic stress induces an early and massive, at least in part apoptotic, cellular death, specifically in the BV2 cell line.

### 3.2. The IRE1α-sXbp1 Arm of the UPR Is Preferentially Activated in N2a Cells but Not in BV2 Cells Under Proteotoxic Stress

Extensive evidence, including our previous studies, demonstrates that proteasome inhibition induces ER stress. However, the profile of UPR activation may vary depending on cell type, age, and the stressor agent, ultimately determining the cell fate [28]. Thus, we next analyzed the profile of activation of the UPR induced by proteasome inhibition in both cell lines. For simplicity, and based on previous works [29], we selected specific markers of the three UPR arms.

The expression of the chaperone Grp78 was chosen for the ATF6α arm. As shown (Figure 2A), proteasome inhibition induced an early and significant transcriptional up-regulation of this gene. Similarly, the protein content increased in a time-dependent manner, becoming significantly different from control cells at 6 h in both cell lines. The activation of the IRE1-α arm was assessed by quantifying the amount of spliced xbp1 mRNA. This mRNA is processed by the endoribonuclease activity of IRE1-α, which catalyzes the removal of an unconventional 26-nucleotide intron, resulting in a spliced mRNA (sxbp1), which codes for the pro-survival transcription factor sXbp1 [30]. The quantity of sxbp1 mRNA exhibited an early and significant increase in N2a cells, while it was weakly induced or even decreased during the first hour in BV2 cells. Similarly, the content of the sXbp1 protein increased significantly in a time-dependent manner in N2a cells but did not show any significant change in expression in BV2 cells (Figure 2B). Finally, the PERK pathway was assessed through the expression of the pro-apoptotic transcription factor CHOP. PERK activation leads to the phosphorylation of the α-subunit of the eukaryotic initiation factor 2 (eIF2α) in the cytoplasm, attenuating protein translation. However, in this situation, specific mRNAs such as ATF4 and CHOP are preferentially synthesized and translated [29]. Proteasome inhibition significantly induced the transcriptional expression of chop in both cell lines. However, the amount of CHOP protein was early increased in BV2 compared with N2a cells. CHOP expression was significantly elevated, compared with control situations, from 2 to 6 h in BV2 cells, and later, from 4 to 6 h in N2a cells (Figure 2C). Taken together, these data indicate that proteotoxic stress induced ER stress in both cell lines. However, both the kinetics and intensity of UPR activation varied between the two cell lines. N2a cells displayed a canonical UPR activation characterized by a predominance of the IRE1α-sXbp1 pathway, whereas BV2 cells did not induce the IRE1α-sXbp1 arm and displayed a preference for activating the PERK-CHOP pathway. The transcription factors CHOP and sXbp1 are associated with cell death and cell survival, respectively [28,31,32,33]. Consequently, the differences observed in the kinetics and intensity of UPR activation in both cell lines were associated with cell fate.

As illustrated in Figure 3A, the CHOP/sXbp1 protein ratio showed an inverse correlation with cell survival in both cell lines. Moreover, direct induction of ER stress through incubation with tunicamycin led to complete UPR activation in both cell lines, exhibiting similar kinetics, chop/sxbp1 mRNA ratio, and no differences in cell survival (Appendix A). Finally, the simultaneous inhibition of both IRE1α and PERK pathways using specific inhibitors (STF-083010 and GSK2606414, respectively) significantly increased or decreased cellular viability in BV2 or N2a cells, respectively (Figure 3B). Thus, these data support the notion that the distinct vulnerability observed in our model of proteotoxic stress in the BV2 and N2a cell lines might be determined, at least in part, by the different kinetics and/or profile of UPR activation.

### 3.3. Proteotoxic Stress-Induced Autophagy Activation in Both Cell Lines but with Different Kinetics

We and others have previously demonstrated that autophagy activation in response to proteasome inhibition is crucial for proteostasis restoration both in vitro and in vivo [9,10]. However, it is unknown whether this functional cooperation between the proteasome and autophagy is similar for all cell types. To address this question, we investigated autophagy activation in both cell lines following proteasome inhibition. First, basal autophagy was evaluated by analyzing autophagic flux using bafilomycin and chloroquine. Bafilomycin treatment led to higher LC3-II accumulation in BV2 compared with N2a cells, indicating a faster basal autophagic flux in BV2 cells. This was also qualitatively supported by staining experiments with acridine orange, which labels acidic compartments, most probably corresponding to lysosomes (Appendix A). Next, we investigated whether proteotoxic stress induced autophagy. As shown in Figure 4A, BV2 and N2a cells stimulated basal autophagic activity in response to proteasome inhibition, but with different kinetics. The amount of LC3-II showed a faster and more significant increase during the first 2 h (peaking at 1 h) in N2a cells, returning then to the basal level. However, BV2 cells exhibited a later and significant increase in the basal content of LC3-II during the first 4 h (peaking at 4 h), subsequently returning to the basal situation. These data support cell-specific differences in the kinetics of autophagy activation in response to proteotoxic stress in N2a and BV2 cells.

### 3.4. Proteotoxic Stress Regulated the Adaptor Protein p62/SQSTM1 in a Cell-Specific Manner

To go deeper, we analyzed the expression of the adaptor protein p62/SQSTM1 (p62 hereafter). p62 is involved in the cargo of aggregated ubiquitinated proteins into the autophagosome [34,35], being incorporated into the autophagosome to be finally degraded with LC3-II and the cargo material. Indeed, p62 degradation could also be considered a marker of the autophagic flux because, in general, when autophagy is inhibited, p62 accumulates, but it tends to decrease when it is activated [36,37]. As shown in Figure 4B, proteotoxic stress induced a significant and cell-specific transcriptional up-regulation of p62 in BV2 but not in N2a cells. Importantly, the amount of p62 increased significantly in both cell types but only accumulated in BV2 cells (Figure 4C), despite these cells displaying a higher basal autophagic flux (Appendix A).

These data could suggest a defective autophagy resolution in BV2, compared with N2a cells, under proteotoxic stress. However, it has been shown that proteasome inhibition stimulates the phosphorylation of p62 at its ubiquitin-association domain, increasing its affinity to ubiquitinated proteins and leading to the incorporation of p62 and the cargo material into the autophagosomes [38,39]. Thus, we analyzed the level of phospho-(S405)-p62 in both cell lines following proteasome inhibition. As shown (Figure 4C; middle panel), BV2 cells did not exhibit a significant alteration in the amount of phospho-(S405)-p62 compared with the control condition. On the contrary, N2a cells showed a significant and sustained elevation in the level of phospho-(S405)-p62 (Figure 4D), indicating also the existence of cell-specific post-translational regulation of p62 between BV2 and N2a cells. Thus, the lower amount of phospho-(S405)-p62 could also contribute to p62 accumulation because of reduced incorporation into the autophagosomes. To test this possibility, we analyzed the LC3-II/phospho-(S405)-p62 ratio under proteotoxic stress. If phospho-(S405)-p62 and LC3-II are concurrently degraded in the autophagolysosome, accumulation of both proteins should occur when autophagosome degradation is inhibited. As illustrated in Figure 4E, proteotoxic stress did not modify the cell-specific autophagic flux, remaining higher in BV2 than in N2a cells. Importantly, the inhibition of autophagosome degradation with bafilomycin during proteotoxic stress increased the ratio of LC3-II/phospho-(S405)-p62, compared with MG132 alone, in BV2 cells (Figure 4F). This effect was attributed to a preferential accumulation of LC3-II rather than of phospho-(S405)-p62, suggesting an enrichment of autophagolysosomes lacking p62. However, in N2a cells, this ratio was similar in both situations (MG132 and MG + bafilomycin), supporting the simultaneous degradation of both proteins within the autophagolysosomes. Thus, the reduced content of phospho-(S405)-p62 in BV2 cells appears to be primarily attributed to diminished p62 phosphorylation rather than accelerated autophagic degradation. Collectively, these data demonstrated a cell-specific regulation of p62 in BV2 (transcriptional and post-translational) and N2a cells (post-translational) under proteotoxic stress.

### 3.5. Autophagy Activation Induced by Proteotoxic Stress Restored Cellular Proteostasis in N2a but Not in BV2 Cells

As previously mentioned, the phosphorylation of p62 at S405 enhances its affinity to ubiquitinated proteins, and this post-translational modification is essential for the selective degradation of p62-cargo by autophagy under proteotoxic stress [38,39]. Thus, we investigated whether cell-specific regulation of p62 affected cellular proteostasis. To address this issue, we analyzed the accumulation of polyubiquitinated proteins in both cell lines following proteasome inhibition. Remarkably, despite autophagy being activated in both cell lines, polyubiquitinated proteins, especially those with high molecular weight, exhibited a significant time-dependent accumulation in BV2 but not in N2a cells (Figure 5A,B). These data strongly support the notion that cell-specific post-translational regulation of p62 has functional consequences on cellular proteostasis. Particularly, the interplay between the proteasome and autophagy for restoring cellular proteostasis appears to be more efficient in N2a compared with BV2 cells. Indeed, autophagy inhibition significantly decreased cell viability induced by proteotoxic stress in N2a cells but significantly increased it in BV2 cells (Figure 5C). Thus, these data suggest that in our model of proteotoxic stress, autophagy activation in BV2 cells might be primarily involved in functions other than proteostasis restoration.

### 3.6. Autophagy-Regulated Phagocytic Activity in Stressed BV2 Cells

Considering that microglial cells are the most important phagocytic immune cells in the brain, and previous studies have shown that autophagy regulates phagocytosis in microglial cells under various challenges [14], we explored the possibility that microglial autophagy might be regulating phagocytic activity rather than contributing to proteostasis restoration. For that, we performed a phagocytosis assay using BV2 cell debris labeled with the red fluorescent 5-(and-6)-carboxytetramethylrhodamine succinimidyl ester (5(6)-TAMRA) as target cells. TAMRA-labelled BV2 cell debris was added to BV2 cell cultures, and phagocytic activity was evaluated by flow cytometry, quantifying the proportion of BV2 cells containing red fluorescence (Figure 5D). As shown in Figure 5E, the phagocytic activity was stimulated in the presence of TAMRA-labeled BV2 cell debris and appeared to be independent of autophagy activity when cells were not stressed. Nevertheless, proteasome inhibition markedly increased phagocytosis in BV2 cells, indicating that proteasome activity somehow regulates microglial phagocytosis. Importantly, under these conditions, previous inhibition of autophagy induction with 3-MA significantly decreased phagocytic activity, indicating that autophagy actively regulates phagocytosis in BV2 microglial cells. These results suggest that the induction of microglial autophagy by proteotoxic stress may contribute to the regulation of phagocytosis rather than solely maintaining cellular proteostasis.

### 3.7. Proteotoxic Stress Activated Cell-Specific Pathways in N2a and BV2 Cells

Given the divergent autophagic responses to proteotoxic stress in both cell lines, we subsequently investigated the activation state of key signaling pathways that trigger and/or participate in autophagy regulation. As depicted in Figure 6A, proteasome inhibition led to a significant reduction in the levels of phospho-(S2448)-mTORC1 in both cell lines. Notably, this effect occurred early in N2a cells, showing sustained activity from 1 to 6 h. In contrast, in BV2 cells it was delayed and exhibited a transient profile (from 2 to 4 h). However, recent data suggest that mTORC1 dephosphorylation at S2448 might not necessarily imply its inactivation [40]. To clarify this, we assessed the phosphorylation level of ULK1, a downstream target of mTORC1 involved in autophagy activation. The phosphorylation of ULK1 at S757, primarily carried out by the protein kinase mTORC1, was significantly reduced in both cell lines just 6 h after proteasome inhibition, later than the timeframe of autophagy activation (Figure 6A). Similar results were obtained with the mTORC1 downstream protein ribosomal protein S6 kinase. Thus, these data support that mTORC1 might not be a crucial regulator of autophagy activation in our model of proteotoxic stress.

We also analyzed the Ser/Thr protein kinase Akt. This protein is primarily regulated by phosphorylation at S473 by mTORC2 [41] and at T308 by the phosphoinositide-dependent protein kinase 1 (PDK1) [42]. Moreover, Akt can directly phosphorylate mTOR [43] and its regulators TSC1/2, modulating autophagy activity [44,45]. Interestingly, Akt displayed differential phosphorylation in both cell lines under proteotoxic stress. As illustrated in Figure 6B, BV2 cells significantly increased the content of phospho-(S473)-Akt during the first two hours but significantly decreased that of phospho-(T308)-Akt from 4 to 6 h. In contrast, N2a cells did not alter the content of phospho-(S473)-Akt but significantly increased that of phospho-(T308)-Akt. Importantly, the pattern of Akt phosphorylation plays a crucial role in specifying Akt targets. More precisely, Akt phosphorylated exclusively at T308 can target GSK-3β and TSC2, but not FOXO1/FOXO3. In contrast, the phosphorylation of FOXO1/FOXO3 by Akt requires phosphorylation in S473 alone or in conjunction with T308 [46,47]. As GSK-3β, FOXO1, and FOXO3 play pivotal roles in autophagy activation [7,26,48,49,50,51,52], we subsequently examined the phosphorylation status of these three downstream targets of Akt in both cell lines. As illustrated in Figure 7A, the kinase GSK-3β exhibited a cell-specific response, with BV2 cells showing a significant reduction in the phospho-(S9)-GSK-3β/GSK-3β ratio, suggesting sustained GSK-3β activity throughout the assessed time. In contrast, N2a cells displayed a transient and significant increase in the phospho-(S9)-GSK-3β/GSK-3β ratio, indicating early and transient inactivation of GSK-3β.

On the other hand, the Forkhead Box O transcription factors are involved in controlling various cellular functions, including autophagy flux [53,54]. First, we analyzed the transcriptional expression of different foxo genes, such as foxo1, foxo3, and foxo6, in both cell lines (Appendix A). The foxo3 gene exhibited the highest expression in both cell lines, with similar levels in BV2 and N2a cells. However, foxo6 was barely detected in both cell lines, and foxo1 was predominantly expressed in BV2 cells. Thus, foxo genes are differentially expressed in microglial and neuronal cell lines, with a predominance of foxo3 in N2a and foxo1 and foxo3 in BV2 cells. So, we analyzed the expression of FOXO1 and FOXO3 proteins. As shown (Figure 7B,C), proteotoxic stress differentially affected FOXO expression in both cell lines. Phosphorylation of FOXO1 decreased in a time-dependent manner in both cell lines. However, the amount of FOXO1 remained higher in BV2 cells, probably because of its transcriptional up-regulation (see also Appendix A). However, FOXO1 decreased in a time-dependent manner in N2a cells. On the contrary, FOXO3 was predominantly phosphorylated in BV2 cells, resulting in a significant reduction, whereas N2a cells significantly increased the content of FOXO3. Collectively, these findings agree with the cell-specific Akt phosphorylation profile observed in BV2 and N2a cells. Furthermore, they emphasize the cell-specific activation of FOXO in response to proteotoxic stress in both microglial and neuronal cell lines, suggesting a higher FOXO1/FOXO3 ratio in BV2 cells but a FOXO3/FOXO1 ratio in N2a cells. Finally, we assessed the β-catenin protein. The interplay between Wnt/β-catenin and autophagy has been well-documented [55]. As illustrated in Figure 8, the response to proteotoxic stress differed between both cell lines. BV2 cells exhibited a significant time-dependent accumulation of β-catenin, suggesting that proteotoxic stress activated this protein in BV2 cells. However, this phenomenon was not observed in N2a cells. Notably, the dynamic of β-catenin phosphorylation was also cell-specific. Proteotoxic stress led to a significant time-dependent increase in β-catenin phosphorylation in N2a cells. However, in BV2 cells, there was an initial significant increase in phosphorylated β-catenin, followed by a subsequent decrease to basal levels (Figure 8; upper panel). Given that phosphorylation is a prerequisite for β-catenin degradation, these results suggest a preferential activation of β-catenin in BV2 cells compared with N2a.

To go deeper, we further assessed the expression of two genes regulated by the Wnt/β-catenin pathway, namely the vascular endothelial growth factor (Vegf) [56,57,58] and the interleukin-6 (IL-6) [57]. Importantly, proteotoxic stress elicited a significant transcriptional up-regulation of both genes exclusively in BV2 cells, providing indirect evidence for the activation of the Wnt/β-catenin pathway (Appendix A). Furthermore, it is noteworthy that IL-6 expression was specific in this context because the expression of other pro-inflammatory cytokines, such as TNF-α and IL-1β, was significantly reduced when compared with the control condition (see Appendix A).

## 4. Discussion

In the present study, we investigated the interplay between the proteasome, UPR, and autophagy in two CNS-derived cell lines. Our key findings can be summarized as follows: (i) Microglial cells demonstrated heightened susceptibility to proteotoxic stress in comparison to neurons; (ii) UPR activation in microglial cells revealed a predominantly pro-apoptotic profile, contrasting with the pro-survival pattern observed in neuronal cells; (iii) proteotoxic stress led to autophagic activation in both cell lines, but with distinct functional consequences. In neuronal cells, autophagy played a crucial role in restoring proteostasis, whereas in microglial cells, autophagy primarily regulated phagocytosis; and (iv) in microglial cells, proteotoxic stress predominantly activated the mTORC2-Akt-GSK3β-FOXO1-β-catenin pathway, whereas neurons activated the PDK1-Akt-FOXO3 pathway.

Our findings demonstrate that maintaining proteostasis in microglial cells predominantly relies on proteasome activity, whereas neurons depend on both proteolytic systems. In this regard, it has been shown that neuronal disruption of the 26S proteasome [59] or autophagy [11,12] resulted in profound neurodegeneration and the formation of inclusion bodies, even in the absence of additional stress. By contrast, autophagy disruption in microglia did not induce microglial degeneration [60,61]. Moreover, astrocytes exhibited higher proteasome activity compared with neurons [62]. Collectively, these data strongly suggest that glial cells seem to be more susceptible to proteasome disruption than neurons. Interestingly, proteasome inhibition induced ER stress in both cell types, leading to UPR activation but with notable cell-specific differences. Microglial cells showed an early and prominent activation of the pro-apoptotic pathway PERK-CHOP, while neurons activated the pro-survival pathway IRE1α-sXbp1. Importantly, our findings coincide with observations reported by [15], in both primary murine microglia cultures and BV2 cells. Specifically, they showed significant activation of the PERK-CHOP pathway, minimal or weak transcriptional activation of the IRE1α-sXbp1 arm (with no analysis of sXbp1 protein), and accumulation of polyubiquitinated proteins. This distinctive molecular response might be responsible for the higher sensitivity of microglial cells to proteasome inhibition, as we corroborated by both cell viability assays and UPR inhibition. Importantly, similar results to those observed in cell lines have been obtained in preliminary studies in primary cultures of microglial and neuronal cells. Specifically, microglial cells are more sensitive to proteasome inhibition, activate the PERK-CHOP pathway preferentially, accumulate higher amounts of high-molecular-weight polyubiquitinated proteins, and activate the β-catenin pathway compared with neurons (see Appendix A). Proper activation of the IRE1α-sXbp1 pathway is critical for maintaining cell viability under stress situations in different cell types. For instance, the loss of function in the sXbp1 branch increased protein stress toxicity in a human tautopathy model in *Caenorhabditis elegans* neurons, while its constitutive activation significantly ameliorated the pathology [63]. Similarly, sustained activation of IRE1α in melanoma cells enhanced survival against ER stress-induced apoptosis [64]. Conversely, the inhibition of IRE1α and the subsequent reduction in spliced Xbp1 protein increased mortality in anti-estrogen treatment for breast cancer cells [65]. The mechanism governing the transition between survival and apoptosis in response to ER stress is not fully understood. It may not solely depend on the vigor of activation in one UPR branch over the other; it could also be linked to the relative promptness of activation of both pathways (IRE1α-sXbp1 and PERK-CHOP), as previously demonstrated by [66]. Thus, the different kinetics of activation of the IRE1α-sXbp1 and PERK-CHOP pathways observed in microglial cells and neurons could be one of the factors determining the survival/death dichotomy.

Our results have also revealed that proteasome inhibition activated autophagy in both cell types. These findings are consistent with previous in vitro and in vivo studies [7,9] and reinforce the concept of functional cooperation between both protein degradation systems. However, most studies on the role of autophagy in cellular proteostasis have been performed in neurons, while little is known about this role in microglial cells. We demonstrated that microglial cells have faster autophagic flux than neurons. Both cell lines stimulated autophagy in response to proteotoxic stress but to fulfill distinct cellular functions. Neurons did not show accumulation of high-molecular-weight polyubiquitinated proteins, p62, nor β-catenin, suggesting that autophagy is degrading intracellularly accumulated material to restore neuronal proteostasis. By contrast, despite microglial cells displaying higher autophagic flux, they showed a time-dependent accumulation of high-molecular-weight polyubiquitinated proteins, as well as p62 and β-catenin, supporting a limited role of autophagy in restoring microglial proteostasis. This cell-specific difference could be attributed, at least in part, to the different post-translational regulation of the p62 protein (phosphorylated at S405 in neurons but not in microglia cells). However, we cannot rule out the possibility that other factors could participate in this process. Additionally, we demonstrated that inhibition of autophagy in stressed microglial cells resulted in a significant reduction in phagocytic activity. In this regard, recent in vivo studies have underscored the interaction in microglial cells between autophagy activation and phagocytic degradation of α-synuclein or Aβ peptide released from neurons [25,27,67], as well as myelin [24]. In all the cases, autophagy inhibition drastically diminished the degradation of the phagocytosed material. Thus, we propose that microglial autophagy is primarily directed toward the degradation of externally incorporated material through phagocytosis, a phenomenon known as LAP [68]. In our case, the phagocytosed material could be the BV2 cells undergoing apoptosis because of the pro-apoptotic profile of UPR activation, as previously shown for macrophages [69]. Importantly, neuronal-released α-synuclein induced microglial transcriptional up-regulation of p62, but not lc3, via the TLR4-NF-κB pathway, as well as p62 protein overexpression and accumulation [25], similarly to what we observed in our model of proteotoxic stress in microglial BV2 cells. However, the phosphorylation state of p62 was not analyzed in that work. Thus, we speculate that p62 might be a key player in regulating autophagy or phagocytosis depending on its phosphorylation state, an issue that needs to be investigated.

The different role of autophagy in BV2 and N2a cells was also paralleled by cell-specific signaling signatures. In this regard, BV2 cells activated the mTORC2-Akt-FOXO1-β-catenin pathway, while N2a cells stimulated the PDK1-Akt-FOXO3 axis. Importantly, our results strongly suggest that Akt could potentially serve as a pivotal branching point, facilitating the differential regulation of downstream substrates and influencing the functions signaled by the different pathways. Indeed, we demonstrated preferential phosphorylation of FOXO3 in BV2 cells, but not in N2a cells, which is compatible with the cell-specific phosphorylation of Akt in S473 [46,47]. Consequently, BV2 cells predominantly activated FOXO1, whereas N2a cells activated FOXO3. Interestingly, activation of the phospho-Akt-Ser473-FOXO1 pathway has been previously described in microglial cells after lithium treatment [70], pointing to the phospho-Akt-Ser473-FOXO1 pathway as a critical element regulating cellular physiology in BV2 cells [71]. Moreover, FOXO1 has been shown to regulate autophagy flux in cultured human QBC939 cells [72], suggesting that FOXO1 could be involved in regulating autophagic flux in microglial BV2 cells.

Another cell-specific difference was the activation of β-catenin in BV2 but not in N2a cells. β-catenin is a highly regulated protein with involvement in many cellular functions. One of the factors regulating β-catenin is FOXO3, which can block β-catenin action and expression by different mechanisms [73,74]. For example, overexpression of FOXO3 downregulated the expression of β-catenin target genes [73]. Here, we demonstrated that BV2 overexpressed β-catenin but not FOXO3, while N2a overexpressed FOXO3 but not β-catenin. Taken together, these data allow us to speculate about the possibility that under proteotoxic stress, FOXO3 could somehow be regulating the expression of β-catenin in BV2 and N2a cells. An outstanding issue is unraveling the molecular mechanisms that dictate the cell-specific role of autophagy under proteotoxic stress. Importantly, FOXO1 has been described to regulate phagocytosis in macrophages [75], dendritic cells [76], and neutrophils [77]. However, data in microglial cells are lacking. We provide evidence supporting FOXO1 as a potential molecular factor regulating phagocytosis in microglial BV2 cells, supporting a relevant and common role for FOXO1 in phagocytic cells. On the other hand, the predominant expression of FOXO3 in neuronal N2a cells could be underlying, among other factors, canonical autophagy activation as previously shown in the neuronal HT22 cell line under oxidative stress [78], ischemic stroke [79], and in a mouse model of Huntington disease [80].

In summary, we provide solid evidence indicating that microglial cells and neurons displayed cell-specific responses to proteotoxic stress, highlighting the different roles of autophagy in both cell types. We also propose that cell-specific Akt and p62 phosphorylation could be involved in determining cellular responses to proteotoxic stress (see Figure 9 for a summary).

## 5. Conclusions

In conclusion, this study reveals that microglial and neuronal cells respond differently to proteotoxic stress. Microglial cells show a primary reliance on autophagy to regulate phagocytic activity under stress, rather than to restore cellular proteostasis, which contrasts with the neuronal response. Neurons, on the other hand, effectively use autophagy to maintain proteostasis, making them more resilient to proteotoxic stress. These cell-specific responses underscore distinct autophagy roles in the central nervous system, shedding light on the molecular mechanisms that may guide therapeutic approaches for neurodegenerative diseases associated with proteotoxic stress.

## Figures and Tables

**Figure 1 cells-13-02069-f001:**
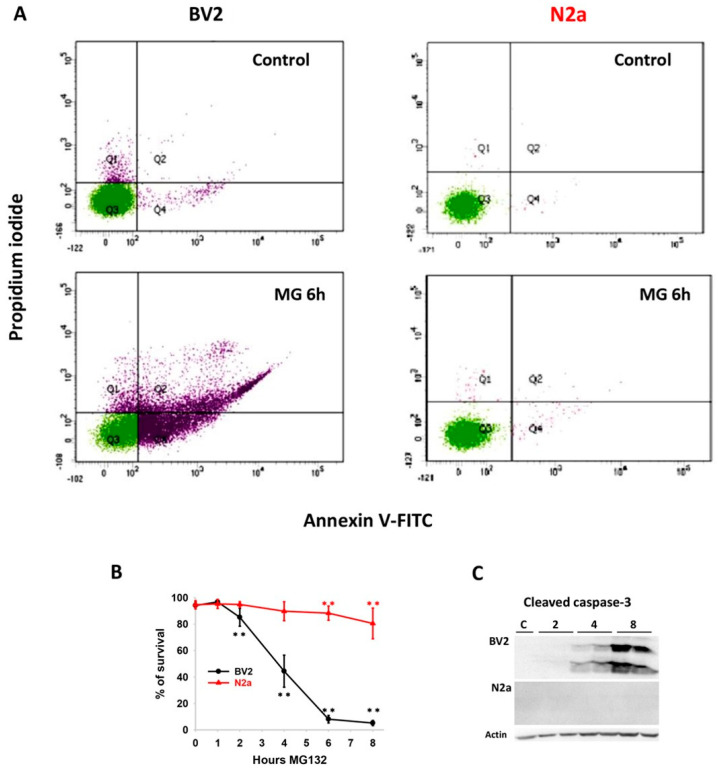
**Analysis of cell viability and apoptosis following proteasome inhibition**. (**A**) Measure of apoptosis by flow cytometry after 6 h of MG132 treatment, using the combination of Annexin V-FITC/propidium iodide staining in BV2 and N2a cells. (**B**) Analysis of cell survival following 1, 2, 4, 6, and 8 h of MG132 incubation in both cell lines. (**C**) Representative image of the western blot of cleaved caspase-3 at the different time points. Bands correspond to 17 and 19 kDa cleaved fragments. No expression detected in N2a cells. Data are expressed as means of the percentage of cell survival ± SD. Cell viability assay was performed at least four times. Statistical significance ** *p* < 0.01.

**Figure 2 cells-13-02069-f002:**
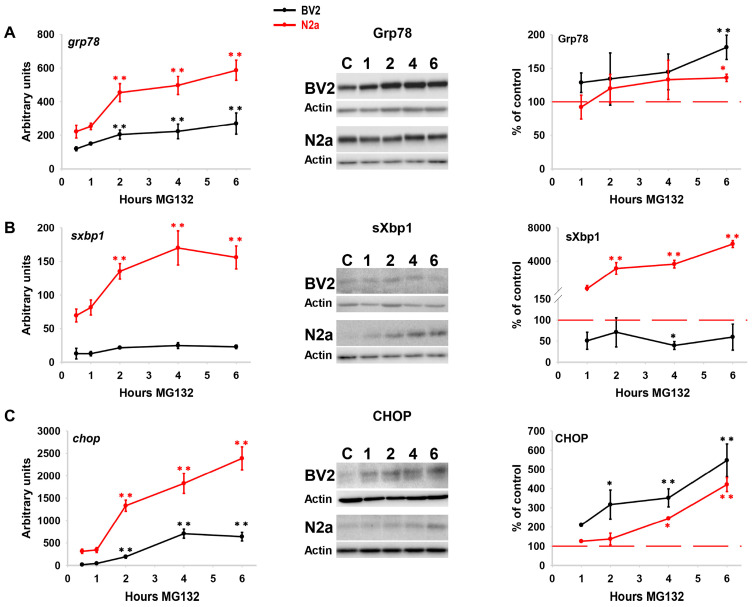
**UPR markers expression**. Quantification of the transcriptional expression (left), representative western blots (middle), and protein levels (right) of the UPR downstream markers Grp78 (**A**), sXbp1 (**B**), and CHOP (**C**), following MG132 treatment at different time points in both cell lines. Actin is included as loading control. Data are expressed as arbitrary units of fold change in the gene expression and as mean of the percentage of the optical density (OD) normalized to control for the protein expression. Experiments were repeated at least three times. Statistical significance * *p* < 0.05 and ** *p* < 0.01.

**Figure 3 cells-13-02069-f003:**
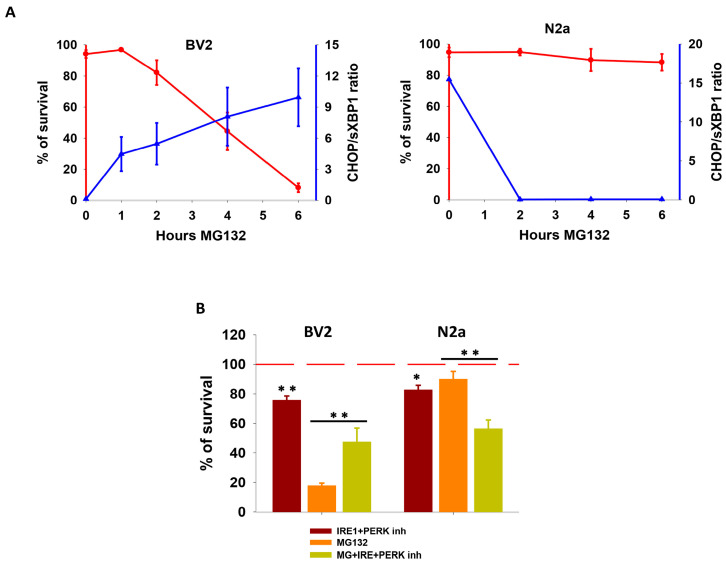
**Assessment of the implication of IRE1α and PERK branches on cell survival**. (**A**) Correlation between the percentage of cell survival and the protein CHOP/sXBP1 ratio in BV2 (left) and N2a (right) cells following MG132 treatment. (**B**) Analysis of cell survival following 6 h of simultaneous inhibition of IRE1α and PERK in presence and absence of MG132. Data are expressed as mean of the percentage of cell survival or the ratio of OD ± SD. Experiments of cell survival were repeated at least five times. Statistical significance * *p* < 0.05 and ** *p* < 0.01.

**Figure 4 cells-13-02069-f004:**
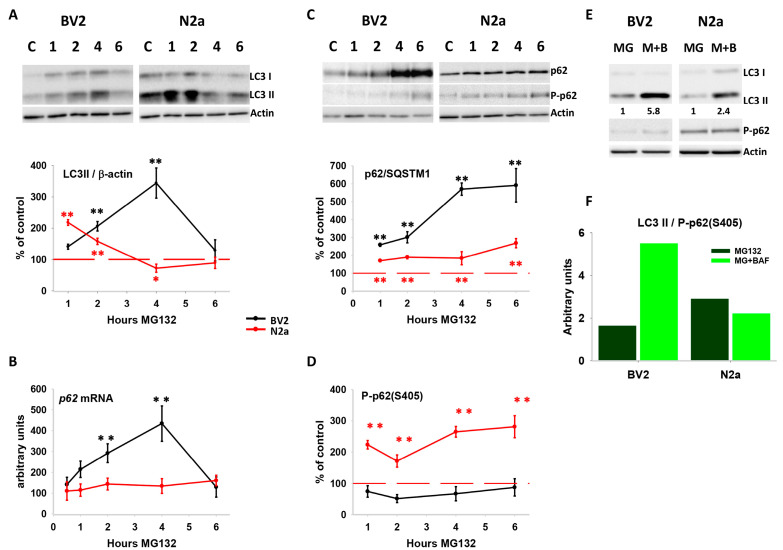
Evaluation of autophagic markers following proteasome inhibition in both cell lines. (**A**) Representative images of western blots of the autophagic marker LC3-II (and actin as loading control) following MG132 treatment at different time points (upper panel). Quantification of LC3-II expression normalized to each control (lower panel). (**B**) Quantification of the transcriptional expression of p62 gene following proteasome inhibition. (**C**) Representative images of western blot of p62 (upper panel), P-p62-(S405) (middle panel), and actin as loading control after different time points of MG132 incubation. Below is shown the quantification of the percentage of p62 protein level (normalized to control) after different times of MG132 treatment. (**D**) The same for P-p62-(S405). (**E**) Representative western blot of LC3-II (upper panel), P-p62(S405) (middle panel), and actin as loading control, in the presence of MG132 (MG) and MG132 + bafilomycin (M + B). Relative intensity of the LC3-II band is indicated at the bottom of the gel, normalized to MG condition. (**F**) Quantification of the LC3-II/P-p62-(S405) ratio after 5 h of MG132 incubation, with or without bafilomycin in BV2 and N2a cell lines. Data are expressed as (i) means of the percentage of the optical density (OD) normalized to control for the protein expression; (ii) means of the arbitrary units of fold change in the gene expression; and (iii) means of the LC3-II/P-p62-(S405) ratio. Experiments were repeated at least three times. Statistical significance * *p* < 0.05 and ** *p* < 0.01.

**Figure 5 cells-13-02069-f005:**
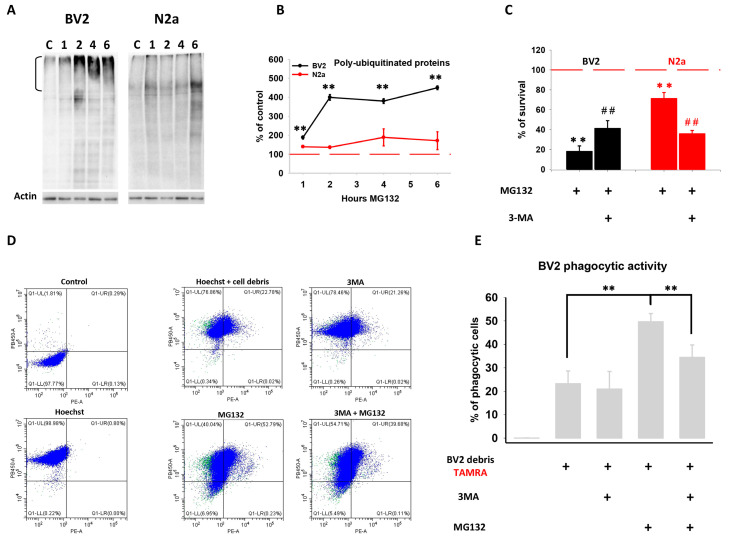
Analysis of polyubiquitinated protein accumulation and the effect of autophagy inhibition on cell survival and phagocytic activity following MG132 treatment. (**A**) Representative western blot of accumulated polyubiquitinated proteins following proteasome inhibition. (**B**) Quantification of the total amount of polyubiquitinated proteins showing accumulation in BV2 but not in N2a cells. Note the absence of high-molecular-weight polyubiquitinated proteins within N2a cells. (**C**) Quantification of the percentage of cell survival following the combined treatment with MG132 and the autophagy inhibitor 3-MA. (**D**) Representative scatter plot of flow cytometry analysis in BV2 cells. The *Y*-axis (PB450) corresponds to Hoechst staining, and the *X*-axis corresponds to 734 TAMRA staining. The upper plots are non-stained (NS) cells (left) and cells stained only with Hoechst (right). The upper right quadrant represents phagocytic cells (positive for both Hoechst and TAMRA). (**E**) Quantification of the percentage of phagocytic cells treated with MG132, 3-MA, or the combined treatment for 6 h (3-MA was pre-incubated 1 h before MG132). Experiments were repeated at least three times and five times for the phagocytosis assay. Data are expressed as means of the percentage normalized to control. Statistical significance ** (related to control) or ## (related to MG132 alone) *p* < 0.01.

**Figure 6 cells-13-02069-f006:**
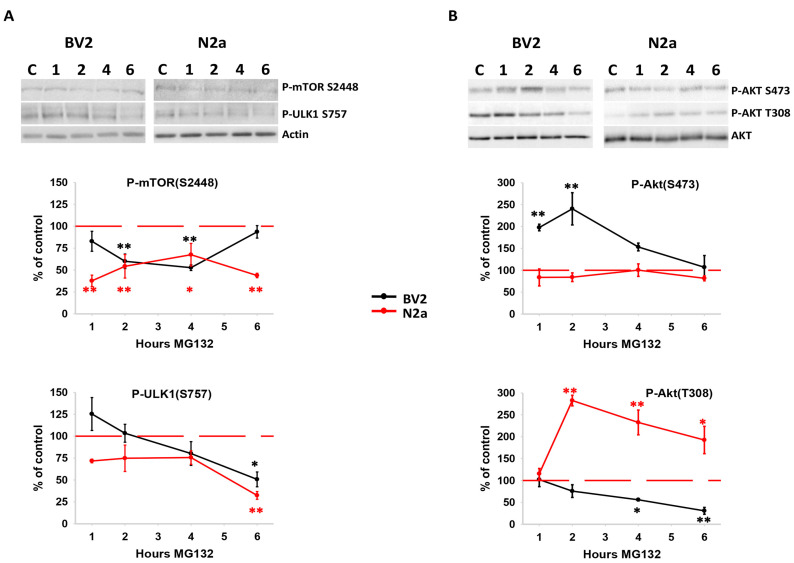
Analysis of mTORC1, ULK1, and Akt phosphorylation levels following proteasome inhibition. (**A**) Representative western blot images of the evolution of P(S2448)-mTOR (upper panel) and P(S757)-ULK1 (middle panel) following proteasome inhibition. Actin was included as loading control. (**B**) Representative western blot images of P(S473)-Akt (upper panel), P(S308)-Akt (middle panel), and total Akt levels (lower panel). Data shown in the graphs are expressed as the mean of the percentage of the optical density (OD) normalized to control. Note the cell-specific difference in the profile of Akt phosphorylation. Experiments were repeated at least three times. Statistical significance * *p* < 0.05 and ** *p* < 0.01.

**Figure 7 cells-13-02069-f007:**
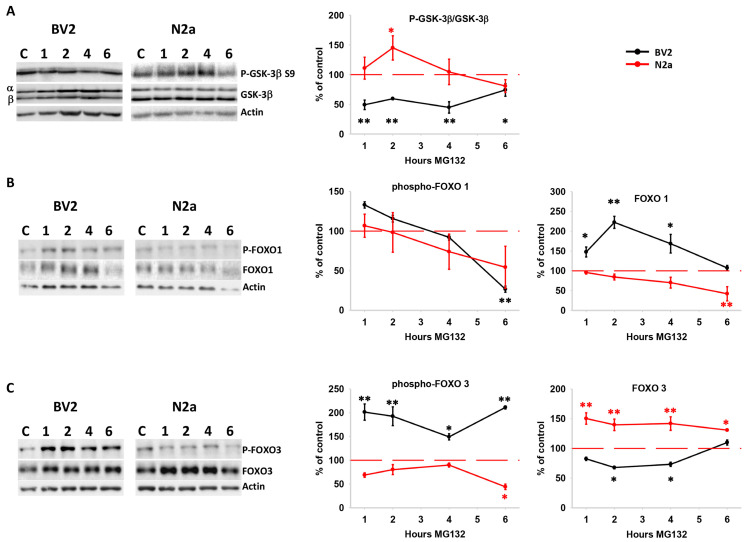
Evaluation of GSK-3β, FOXO1, and FOXO3 phosphorylation levels following proteasome inhibition. (**A**) Representative western blot images of P(S9)-GSK-3β, total GSK-3β, and actin as loading control, following proteasome inhibition. The graph shows the quantification of P(S9)-GSK-3β/GSK-3β ratio. (**B**) Representative western blot images of P(S256)-FOXO1, FOXO1, and graphs showing the quantification of P(S256)-FOXO1 and FOXO1 following proteasome inhibition. (**C**) The same as in (**B**) but for P(S253)-FOXO3 and FOXO3. Actin was included as loading control. The data are expressed as the mean percentage of the ratio of the optical density (OD) normalized to the control. Experiments were repeated at least three times. Statistical significance * *p* < 0.05 and ** *p* < 0.01.

**Figure 8 cells-13-02069-f008:**
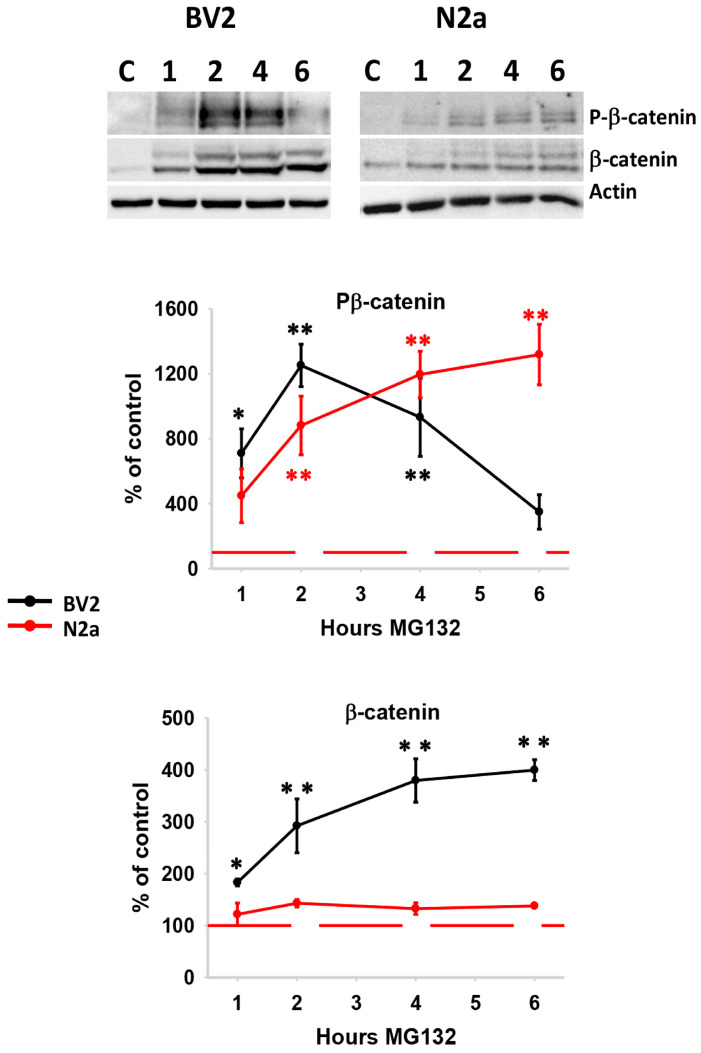
Expression and phosphorylation of β-catenin following proteasome inhibition. Representative western blot images of P(S33/37/T41)-β-catenin and β-catenin following proteasome inhibition. Actin was included as loading control. Quantification of the western blots showing a differential dynamic in the level of both P(S33/37/T41)-β-catenin (middle graph) and β-catenin (lower graph). Experiments were repeated four times. Statistical significance * *p* < 0.05 and ** *p* < 0.01.

**Figure 9 cells-13-02069-f009:**
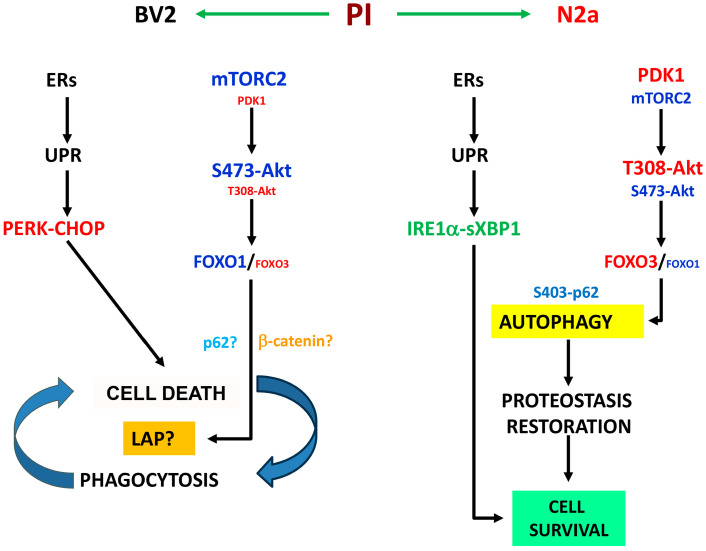
Summary of the cell-specific responses to proteasome inhibition observed in BV2 and N2a cells. In BV2 cells, proteasome inhibition (PI) weakly activates the IRE1α-sXbp1 arm but predominantly triggers the pro-apoptotic PERK-CHOP pathway. Additionally, there is a prominent induction of the mTORC2-AKT-FOXO1-β-catenin pathway, potentially leading to LC3-associated phagocytosis (LAP) and subsequent cell death, which probably stimulates phagocytosis, leading to additional cell death. In N2a cells, PI activates the canonical UPR with predominance of the IRE1α-sXbp1 arm. Simultaneously, it triggers the PDK1-AKT-FOXO3 axis, stimulating autophagy for proteostasis restoration. According to present data, we propose that autophagy appears to play a role in restoring cellular proteostasis in neurons while regulating phagocytosis in microglial cells, likely mediated by differential Akt and p62 phosphorylation levels leading finally to cell survival or cell death.

## Data Availability

The raw data supporting the conclusions of this article will be made available by the authors, without undue reservation.

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
