# Peer review of "Distinct UPR and Autophagic Functions Define Cell-Specific Responses to Proteotoxic Stress in Microglial and Neuronal Cell Lines"

_cells, 2024, doi:10.3390/cells13242069_

Round 1
Reviewer 1 Report
Comments and Suggestions for Authors
This manuscript investigates the effects of proteotoxic stress on two different cell types, with a particular focus on autophagy and the unfolded protein response (UPR) as potential mechanisms. The work presented reflects considerable effort.
However, the manuscript’s overall clarity requires improvement, as the current presentation hinders comprehension. Enhancements in language and structure are needed before acceptance.
1. Title
The title should be revised to include the primary findings, such as the involvement of UPR.
2. Introduction
The introduction would benefit from a stronger reference base, incorporating more relevant literature. Additionally, unrelated content, such as that in lines 54–57, should be removed.
3. Methods
- Clarify the type of control treatment used.
- In the figures, only one control group is presented, despite several time points. Each time point should have its own control.
- Specify the statistical analysis method used for comparisons between control and MG132-treated groups.
- For DMSO treatment, typical concentrations are less than 0.1%. Provide a rationale for using 1% DMSO and evidence that this concentration is non-toxic.
- Notably, the MG132 treatment was applied without FBS, which can independently activate autophagy due to nutrient deprivation. Explain how you controlled for this variable, as the observed autophagy activation may not be solely due to MG132.
- Report the exact concentration of bafilomycin used.
4. Results
- The "control" group is unclear. Define this more explicitly.
- Each time point is independent, so bar charts would be more appropriate than line charts for clearer data visualization.
- In Figure 2, avoid placing the mRNA data in the middle of Western blot bands and the related quantification data.
- For Figure 4E, the experimental design may not fully capture changes in autophagic flux. To assess flux, two groups (B and B+M) are ideal. An increase in LC3-II after M treatment would indicate active autophagic flux.
- For Figures 7B and 7C, consider using p-FOXO1/FOXO1 and p-FOXO3/FOXO3 ratios.
- Some western blot bands are unclear, making it challenging to assess changes. Improving image quality in these figures would enhance clarity.
Author Response
We sincerely thank the reviewer for their valuable suggestions and constructive feedback, which have significantly contributed to improving the quality and clarity of our manuscript. We have carefully considered all the comments and have incorporated most of the suggested changes into the revised version of the manuscript. Below, we provide detailed responses to each point raised and outline the modifications made accordingly. We greatly appreciate your thoughtful review and the opportunity to enhance our work.
Please note that all modifications made in response to the reviewer's comments, except for the replacement of all figures with improved quality, have been highlighted in yellow in the revised version of the manuscript to facilitate the revision process.
The detailed responses to the comments have been attached as a PDF document.

Reviewer 2 Report
Comments and Suggestions for Authors
The manuscript focuses on the interrelationship between proteasomal activity, the UPR, and the autophagy pathway in two CNS-derived cell lines. The manuscript shows powerful and valuable work, with clear and consistent approaches. However, some aspects may be discussed with a deeper insight.
1.- The authors point out that under proteostatic stress the transcript of the chaperone Grp78 is higher in neuronal cell line than in the glial cell line, however, the proteins only increased after 6 hours of treatment and, contrary to what was expected, reached lower levels than the glia, which seems to reveal a different and higher translational regulation in the glial cell line than in the neuronal cell line. A similar situation occurs with CHOP, where the glial line significantly increases its translational expression from the beginning of the treatment, despite the fact that the levels of its transcript are, in the glial cell line, lower than those observed in the neuronal line.
It would be interesting to know the autor´s interpretation of these striking results, which appear to reveal different patterns of gene expression upon inhibition of the proteosomal pathway.
2.- Does autophagy work in the same way in cells which are in permanent division and in those in G0? Could the increase in cell division be understood, under certain conditions, as an auxiliary resource for cellular cleaning?
3.- How do the authors evaluate the fact that their findings have been obtained only in cell lines, whose cell cycle is not the one observed in glia and neurons under natural conditions?
4.- Is it posible to extrapolate these results to cells that are in G0 such as neurons?
It would be interesting to know an alternative approach to understand the implication of these findings closer to those of neurons and microgli
Author Response
We sincerely thank the reviewer for their valuable suggestions and constructive feedback, which have significantly contributed to improving the quality and clarity of our manuscript. We have carefully considered all the comments and have incorporated most of the suggested changes into the revised version of the manuscript. Below, we provide detailed responses to each point raised and outline the modifications made accordingly. We greatly appreciate your thoughtful review and the opportunity to enhance our work.
The detailed responses to the comments have been attached as a PDF document.

Round 2
Reviewer 1 Report
Comments and Suggestions for Authors
The manuscript appears much clearer after the revision. However, despite the authors’ claim that the Western blot images were updated, I could not discern any noticeable differences. Specifically, Figure 2B (sXbp1 for BV2) and Figure 6A (p-mTOR S2448) remain particularly unclear and require further improvement.
Author Response
Comment 1:
The manuscript appears much clearer after the revision. However, despite the authors’ claim that the Western blot images were updated, I could not discern any noticeable differences. Specifically, Figure 2B (sXbp1 for BV2) and Figure 6A (p-mTOR S2448) remain particularly unclear and require further improvement.
Response 1:
Thank you very much for your feedback. We apologize for any confusion and would like to clarify that the incremental enhancements we referred to were made to the overall quality of the JPG figures, not to the individual Western blot images. We agree with the fact that some Western blots present less resolution. The Western blot images were already captured at the highest resolution achievable within our technical limitations. The main limitation is that the expression levels of both sXbp1 and p-mTOR S2448 are inherently low in our experimental model, which unavoidably affects image quality. This necessitates the intensified development and exposure of the corresponding Western blot signals.
Furthermore, the images included in the manuscript's PDF are derived from the JPG files embedded into a Word document and subsequently exported to PDF format, a process that further diminishes resolution. As we are uncertain whether you currently have access to the JPG files of the figures, please inform us if you require any assistance in obtaining them.
Despite these limitations, we believe the Western blot bands are sufficiently clear for accurate interpretation, with the findings effectively represented and supported by the accompanying quantitative graphs. We hope this addresses any concerns.